# An Overview of the Application of Multivariate Analysis to the Evaluation of Beer Sensory Quality and Shelf-Life Stability

**DOI:** 10.3390/foods11142037

**Published:** 2022-07-09

**Authors:** Ana Carolina de Lima, Laura Aceña, Montserrat Mestres, Ricard Boqué

**Affiliations:** 1Chemometrics, Qualimetrics and Nanosensors Group, Department of Analytical Chemistry and Organic Chemistry, Campus Sescelades, Universitat Rovira i Virgili, 43007 Tarragona, Spain; anacarolina.delima@urv.cat; 2Instrumental Sensometry Group (iSens), Department of Analytical Chemistry and Organic Chemistry, Campus Sescelades, Universitat Rovira i Virgili, 43007 Tarragona, Spain; laura.acena@urv.cat (L.A.); montserrat.mestres@urv.cat (M.M.)

**Keywords:** sensory quality, brewing process, aging, multivariate analysis

## Abstract

Achieving beer quality and stability remains the main challenge for the brewing industry. Despite all the technologies available, to obtain a high-quality product, it is important to know and control every step of the beer production process. Since the process has an impact on the quality and stability of the final product, it is important to create mechanisms that help manage and monitor the beer production and aging processes. Multivariate statistical techniques (chemometrics) can be a very useful tool for this purpose, as they facilitate the extraction and interpretation of information from brewing datasets by managing the connections between different types of data with multiple variables. In addition, chemometrics could help to better understand the process and the quality of the product during its shelf life. This review discusses the basis of beer quality and stability and focuses on how chemometrics can be used to monitor and manage the beer quality parameters during the beer production and aging processes.

## 1. Introduction

Alcohol consumption has historically held an important role in social engagement and bonding for many cultures. In Europe, one of the largest alcoholic markets in the world, beer, wine, and spirits are the dominant beverages [1]. In 2021, the main beer producers in Europe: Germany, Poland, Spain, United Kingdom, Netherlands, France, Czech Republic, Romania, and Italy were responsible for producing 75% of the total volume of beer in the continent [2]. 

Most of the market concerning the production of beer in Europe is dominated by a few multinational companies, although in the last decade, there has been an increase in the number of independent craft breweries in countries with no historical links with the tradition of beer crafting, such as Greece, Italy, Portugal and Spain [3,4]. Nowadays, Europe has more than 11,000 active breweries, with around 73% of them being microbreweries [3,4]. The result is that this market has experienced notable changes in the behavior of consumers, who are more aware and demanding in terms of their desired quality and consumption habits [5].

As opposed to industrial beers, craft beers are produced on a small scale by independent craft breweries whose focus is on product innovation, emphasizing flavor, variety of styles, and techniques committed to the sensory quality of the final product [5]. Despite the differences between industrial and craft beers, the basics of the process remain the same, including water, malted barley, hops, and yeast, and producing beer via alcoholic fermentation.

Different factors such as appearance, aroma, taste, and texture define the organoleptic quality of the final product. During the shelf life of the product, the stability of these quality factors can be influenced by several parameters, including raw material, microbial activity, light, oxygen, and temperature [6,7,8]. Therefore, it is important to control both the raw materials and the process variables to achieve the best sensory quality of the beer and ensure its stability, especially in the case of unpasteurized beer. 

Among the different sensory properties of beer, aroma is the most studied [9]. However, most of these studies focus on reviewing the application of multivariate analysis during the aging process, so there is a lack of information on the aroma evolution during the beer production process and also on the evolution of other sensory parameters. That is why this review focuses on the sensory quality of beer but links it to the different brewing stages. In addition, it reviews and describes the use of chemometric methodologies that allow better understanding and monitor different quality parameters during beer production and shelf life. The review is therefore divided into two parts: (*i*) Beer Production Process, focusing on the three main steps that directly influence the final quality of the product: malting, boiling, and fermentation; (*ii*) Analysis of beer quality and stability by means of different chemometric tools, which focuses on exploratory analysis, classification techniques and multivariate calibration. 

## 2. Beer Production Process

The beer production process can be divided into five steps: malting, mashing, boiling, fermentation, and maturation. It starts with malting the barley via controlled steeping, germination and kilning, transforming it into a product that is much more friable, with active and increased enzyme levels and with different chemical and physical properties [10]. 

The next step is to grind the malted barley for mashing. The grinding characteristics (coarse or fine) will depend on the type of equipment used by the brewer. Mashu tuns or lauter tuns use a coarser grind produced by roll mills, but mash filters can use much finer grist produced by a hammer mill [10]. During mashing, the ground malt—and cereal adjuncts, if used—are mixed with hot water and the enzymes present in the cereal will degrade the proteins in small molecules and the starch into small sugars such as maltose and glucose, producing a soluble malt extract, wort. Before being sent to boiling, the wort is separated from the solid part. For the classic infusion mashing, this separation takes place in the mash tun. For the other separation, such as decoction mashing and Double mashing the mash separation is usually carried out using other equipment such as a lauter tun or a mash filter [10]. This filtration process aims to brighten the wort and obtain the maximum amount of extract from the solid residue.

After filtration, the wort is boiled, and the hops are added to the kettle. The main objectives of this step are wort sterilization, extraction of bitter and aroma compounds from hops, coagulation of the excess of proteins and tannins to form trub that can be removed later, color and flavor formation, removal of undesirable volatiles via evaporation, and concentration of the wort by evaporation of water [11,12].

Before sending the wort to the fermentation vessel, it must be aerated and cooled until it reaches the appropriate fermentation temperature, which is typically 8 to 15 °C for lager and 14 to 25 °C for ale beers. During fermentation, yeast is mainly responsible for converting sugars into alcohol and carbon dioxide_,_ but many aroma compounds are also generated. A great fermentation performance demands control of many key variables, such as yeast amount and viability, oxygen input, nutritional wort ability, pH, temperature, and agitation [6,12]. The key to fermentation efficiency and, for definition, of many sensory characteristics that contribute to the quality of the final beer, is the yeast. 

At this point in the brewing process, the wort has already been transformed into beer, known as ‘*green beer*’, and most of the yeast is removed from the vessel, giving a path to maturation. The function of the remaining yeast during maturation is to produce higher carbon dioxide amounts and chemical removal of undesirable compounds. The main objective of this step is to initiate beer clarification and prevent oxidation of the product during the storage by maintaining beer in a reduced state [12]. To contribute to the microbiological stability and help with the clarification process, some types of beer are filtered.

Despite the importance of each step in the beer production process, in this review we will focus on the main three of them: malting, boiling, and fermentation. These steps have important contributions to the quality and stability of the final product, such as improvement of color, transparency, bitterness, and foam properties; wort sterilization and protection; and alcohol, carbon dioxide, and desirable flavors formation. 

### 2.1. Malting

Barley (*Hordeum vulgare* L.) is the main cereal used worldwide for malting due to its high enzymatic ability to convert starch into fermentable sugars [13]. Nevertheless, wheat (*Triticum aestivum* L.) and sorghum (*Sorghum vulgare* L.) are also malted in great quantities, followed by rye (*Secale cereale* L.), oat (*Avena sativum* L.), rice (*Oryza Sativa* L.), and millet (*various* spp.), used in small amounts [14,15,16].

Barley for brewing must meet a series of quality requirements, such as germinative capacity, suitable protein and water contents, sorting (size of kernels), and absence of kernel abnormities and infestation [17,18]. Its composition includes starch, protein, cell wall polysaccharides, and a small amount of fat and minerals. Protein and starch are among the barley components essential for malt and beer quality. According to Jaeger et al. [19] the content of protein in the barley grain is between 8 and 30% of its total mass. Although such a high content of protein is desirable for feeding applications, a lower level is desirable when dealing with malting barley, being 10 to 12% the ideal content of protein. On the contrary, a high level of starch in the barley grain for malting is desirable. Starch is a complex sugar constituted of glucose molecules and its content in a barley grain can range from 45.7% to 70%, which is a suitable amount for malting as, according to Błazewicz et al. [20], barley grain for this purpose should contain 52–67% starch. 

The development and retention of foam in beer depends on the ratio of malt-derived proteins with specific molecular weights, metal cations from malt or water, and natural stabilizers such as polysaccharides derived from malt and iso-α-acids from hops [21,22]. Moreover, fatty acids and basic amino acids derived from malt also affect foam formation and retention [21].

The malting process aims to activate enzymes to promote the barley to provide the optimal levels of saccharides, proteins, free amino nitrogen (FAN), and enzymes to ensure the organoleptic quality of the final product [23]. To initiate the malting process, barley is immersed in the steep water. This phase is called steeping and is designed to increase the moisture level of the grain around 42–47%, to initiate germination and to develop and facilitate the transport and action of the gibberellic acid, an important hormone in the production of enzymes needed by the seed for growth [14,24,25]. The temperature of the water during this phase should be kept between 12–18 °C, to avoid acceleration of the microbial growth or damage to the grain [14,17]. 

According to Brigs et al. [14] an excess of microbes in the steep water is undesirable because they compete with the grain for oxygen and reduce both the percentage and vigor of germination. Furthermore, some microbes could produce mycotoxins such as deoxynivalenol (DON), which damage the yeast and/or are toxic to human beings; others could produce plant growth regulators (including gibberellins), which can inhibit or stimulate the malting process, and some others could produce some agents that cause gushing (over-foam) [14,26]. On the contrary, the controlled microbial activity in the grain results in the production of some hydrolytic enzymes that may improve malt performance during mashing [10,17]. 

When the grain starts the germination process, it is transferred to the germination vessel. [14]. The main goals of this step are the control of the breakdown of cell walls and matrix proteins, production of an optimal level of hydrolytic enzymes, hydrolyzation of certain barley reserves, such as protein, to form free amino nitrogen (FAN), and production of well-modified and balanced green malt for kilning [17].

The physical modifications that occur in the grain during germination are a breakdown of β-glucans and pentosans, followed by the partial degradation of the protein within the cells and breakdown of some of the starch granules [14]. In a greatly modified malt, 90% of β-glucan is broken down. The whole breakdown process is limited by the availability of water. The inhomogeneity of malt modification can cause unexpected problems during brewing, such as slow wort separation, slow beer filtration, and sometimes it can cause haze development. Generally, at this stage, the germination is finished by kilning. 

The kilning stage of the malting process inactivates many microorganisms and stabilizes the grain, allowing long-term storage as it reduces the moisture content of the undried green malt from about 42–48% to 3–6% [27]. The main goals of this stage are to finish the modification process and the growth of the plant to reduce moisture to levels suitable for grain storage, and to develop color and flavor characteristics through the Maillard reactions and, in some cases, caramelization and pyrolysis reactions [17,28]. 

The kilning or roasting process determines the quality of the produced malt and its classification as base or specialty. Base malts provide extracts that are used by the yeast to produce alcohol and flavor compounds, and specialty malts mainly provide color and flavor compounds, adding complexity and diversity to a beer. Table 1 shows the malt type, color description, and general characteristics of the base and specialty malts.

Specialty malts are subject to higher kilning and roast temperatures, which define their colors and flavors. As a result of this process, they lose their enzymatic activity, so they are used in small amounts compared to the base malts, which retain their enzymatic activity [29]. 

### 2.2. Boiling

The main purpose of wort boiling is the evaporation of water and unwanted volatile compounds, isomerization of humulones, fixation of wort composition by inactivation of enzymes, sterilization of wort, and removal of proteins. However, although the boiling stage can positively contribute to the formation of color during beer storage, it should be carefully monitored, as it can lead to the formation of a non-biological haze due to the oxidation of polyphenols derived from malt and hop vegetative matter [28].

*Humulus lupulus*, commonly known as hop, is used in the brewing process during the boiling phase to enhance the quality and stability of beer and to introduce the characteristic hoppy aromas and bitter taste, which leads to desirable flavor for consumers [30,31]. The most important biochemical markers that differentiate hop varieties are hop acids, hop oils, and polyphenols. The hop acids include the so-called α-acids (humulone, cohumulone, and adhumulone) and, β-acids (lupulone, colupulone, and adlupulone) [30,32]. 

The α-acids are tasteless; however, upon boiling in the wort, they are isomerized to the bitter-tasting iso-α-acids or isohumulones [32]. During boiling, only 50% of the α-acids are isomerized and less than 25% of their bittering potential is preserved in beer [33]. This occurs because of the restricted solubility of the α-acids in beer and the slightly acid wort (pH 5–5.5) [31]. On the other hand, the volatile fraction contained in the hop oil (0.5–3% in hops), together with the non-volatile fraction present in the hop polyphenols (3–6%) contribute to a full mouthfeel sensation during beer tasting [31,34]. Therefore, hops play a major role in product quality; in addition, they improve flavor stability, as hop polyphenols have antioxidant capabilities, and their other compounds are effective at masking the development of stale flavors [21]. In order to stabilize the foam head, some breweries use processed hop advanced products, such as reduced iso-α-acids, which enhance foam stability to a greater extent than iso-α-acids [35].

### 2.3. Fermentation

Fermentation is the process in which fermentable sugars are transformed into alcohol, carbon dioxide, and many other compounds by the *Saccharomyces* yeast. To ensure a high-quality product, an effective brewing fermentation is required, as most of the flavor-active compounds of a beer are produced at this stage of the process. 

Although all strains of *Saccharomyces* produce ethanol as an end-product of fermentation, the production of most aroma-active compounds is strictly dependent on the yeast strain chosen for the fermentation and has a great impact on the beer flavor [6,30,36]. 

In beer production, the yeast strains used as starter cultures are divided into two groups: ales and lager yeasts. *Saccharomyces cerevisiae* strains belong to the group of ale yeasts, and require a high range of temperature (14–25 °C), and are often referred to as top-fermenting yeasts because in open fermenters, they rise to the surface of the vessel facilitating their collection [6,30]. On the contrary, lager strains *Saccharomyces pastorianus* are more complex organism than ale yeast, run the fermentation at cool temperatures (8–15 °C), and are known as bottom-fermenting yeasts, because they sediment at the bottom of the vessel at the end of fermentation. 

The key elements produced by yeast, which will determine the final quality of the product, are vicinal diketones (VDKs), esters, and higher alcohols. While esters and higher alcohols could be considered pleasant and desirable volatile constituents in beer, depending on their concentration level, VDKs are frequently considered off-flavors [36]. 

During yeast metabolism, higher alcohols are formed as by-products of amino acid synthesis or catabolism [30,36]. At optimal levels, higher alcohols help with the drinkability and give the beer a desirable and pleasant aroma. On the contrary, above 300 mg/L, these compounds can lead to a strong, pungent smell and taste [30]. 

Esters are one of the most volatile compounds in beer and have a great impact on aroma. In moderate quantities, they add a pleasant and full-bodied character to the aroma [37]. However, in excess, they give an overly fruity aroma, which is considered undesirable by most consumers. 

Finally, oxygen should be also considered, as it plays an essential role in the brewing process, especially during fermentation, where it is required by all yeast cells to support the synthesis of sterols and unsaturated fatty acid components of the cell membranes [14,38]. Oxygen is also required for lipid synthesis, which is necessary to maintain the integrity and function of the plasma membrane and, consequently, the cell replication [39]. On the contrary, an excess of oxygen might damage cell components, contribute to cellular aging, and finally lead to cell death [38,39]. Thus, to obtain a high-quality product, it is necessary to achieve optimum oxygen levels. 

## 3. Analysis of Beer Quality and Stability Using Chemometric Tools

Despite all the technologies available, obtaining a high-quality product is a hard task, as explained above. To achieve this purpose, a brewer must control several parameters in all phases of the process, as described before, ensuring the quality of the final product. In addition, product stability is also essential, as when beer leaves the brewery it is subject to the conditions that distributors and consumers may impose and which can lead to a rapid degradation of quality [6,8,40]. 

Managing and monitoring beer production and aging processes requires both knowledge and techniques to interpret and extract information from datasets with multiple variables [9]. Multivariate statistical techniques, also known as chemometric techniques, are commonly used in food science to help better understand and manage the connections between different types of data. During the beer production process and shelf life of a beer, chemometrics can be a very useful tool for monitoring the quality and stability of the product. 

To monitor beer quality during the beer production process, the most important chemical measurements are those related to sensory perceptions that are routinely measured in a brewery, such as flavor, color, haze, foam, and mouthfeel [41]. On the other hand, monitoring the stability of the product during its shelf life is a hard task due to the complexity of the aging process. During beer aging, many oxidative and non-oxidative reactions occur, in which new molecules may be formed and some existing molecules may increase in concentration or degrade, thus changing the sensory profile of the product [40,42,43,44]. In addition, the chemical and sensorial aspects of the aged beer depend on the beer style and its characteristics (ethanol content, pH, raw materials, and ingredients), the brewing process (temperature and time), and its handling process after packaging (exposition to light, temperature, and vibration) [8,37,40]. 

Many instrumental techniques can be employed to monitor both the beer production process and the aging of the samples, such as nuclear magnetic resonance (NMR) [45,46], near-infrared spectroscopy (NIR) [47,48], electronic-nose (e-nose) [49,50], liquid chromatography–mass spectrometry (HPLC-MS) [13,51], gas chromatography–mass spectrometry (GC-MS) [52,53], and gas chromatography with olfactometric detection (GC-O) [54,55]. In addition to these instrumental techniques, sensory panels can also be performed to better understand the product [56]. In this case, chemometrics can be applied to help explain the relationship between product composition and sensory properties and between sensory properties and instrumental measurements [41]. Table 2 summarizes the scientific literature related to chemometric techniques applied to beer quality and stability over the last 10 years. Other references related to the chemometric techniques used in this article can be found in the References section.

### 3.1. Exploratory Analysis

The chemometric techniques applied to monitor and control the quality of beer production and aging processes cover a wide range of applications, which include exploratory data analysis (EDA), pattern recognition (or classification), and multivariate calibration. The EDA techniques, also known as unsupervised pattern recognition, are usually the first approach used to visualize the data, as they reduce the dimensionality of complex datasets and produce a graphical representation that is easy to understand and interpret. The procedure consists of grouping data based on their similarities, with no prior assumptions about class membership. In food science, the most popular EDA techniques used are principal component analysis (PCA) and hierarchical cluster analysis (HCA) [9]. 

An example of the application of EDA in the beer production process is related to the characterization of raw materials. Inui et al. [70] used PCA to characterize the volatile compounds responsible for the differences in hop aroma characteristics in beer. PCA was able to indicate the positional relationship among six hop aroma characteristics and five hopped beers. Furthermore, the authors could conclude that understanding the relationship between instrumental data and organoleptic evaluation using PCA is effective and reliable for determining the key aroma compounds from numerous unknown components. Dong et al. [52], applied PCA to evaluate the variability of volatile aroma compounds among barley cultivars. The results obtained via PCA analysis showed that the aroma characteristics of brewing barley cultivars from different countries were quite different, while those from the same country were similar, especially the Chinese domestic barley cultivars. In another work, Bettenhausen et al. [23] used PCA to study if there were sensorial differences among six different beers produced from six different malt sources. A descriptive sensory analysis was performed on 45 attributes at 0, 4, and 8 weeks of storage, revealing flavor differences at 8 weeks and thus showing the importance of the malt source in beer flavor stability.

Another example using both EDA techniques, PCA and HCA, was shown by Rendall et al. [53], where the authors analyzed the evolution of the volatile fraction of Portuguese beers over a period of one year under standard shelf storage conditions using gas chromatography coupled with mass spectrometry (GC-MS). A total of 39 lager beers from the same production batch, kept at room temperature for 12 months, were analyzed. The chemometric analysis conducted focused on detecting the early onset of meaningful changes in chemical composition, and then on the analysis and characterization of the evolution of groups of compounds. The chemometric analysis revealed that the chemical composition of the beer presented a statistically significant deviation after 7 months, although the deviation trend had its onset during the sixth month. Furthermore, the authors concluded that there is no single resulting compound that can be identified as a unique aging marker, but rather two sets of compounds acting in a synergistic or antagonist way to produce significant changes in fresh beer flavor. 

To propose a methodology aiming at fast non-destructive metabolomic characterization of a beer, exploring its compositional profile, and highlighting potential trends or peculiar samples, Cavallini et al. [57], combined NMR spectroscopy and chemometrics. In this study, one hundred pale beers from different brands were analyzed. PCA was used for exploratory purposes, on both the full spectrum and features datasets and Multivariate Curve Resolution (MCR) was used for extracting the chemical features from the NMR spectra, which allowed a reduced dataset of resolved relative concentrations to be obtained. This approach using NMR spectroscopy and chemometrics offered clear information about beer composition, providing valuable information about beer characterization that proved to be very useful to the producers in terms of both quality control and innovation.

Coelho et al. [61] studied the beer aging process in wood barrels previously used to age Port wines. The volatile GC-MS fingerprints of unaged beers and beers aged in different times and conditions were analyzed using PCA. Samples showed groups depending on the aging time, which in turn was positively correlated with the presence of a higher number of volatile compounds. Differences in volatile composition were also found between the barrel aged beers and the unaged beer, thus showing that reutilized barrels may have an impact on aged beer production.

### 3.2. Classification Techniques

Pattern recognition methods or classification techniques are supervised methods that aim to recognize patterns in the data and classify observations by assigning a new sample into a category or class [9]. There are two different approaches in classification: discrimination and class modeling. When a discriminant approach is used, the focus is put on the difference between classes, and a sample is always assigned to a given class. For a two-class problem (classes A and B), a sample will be always assigned to A or to B. Examples of discriminant techniques are *k*-nearest neighbors (*k*NN), Artificial Neural Networks (ANN), Linear Discriminant Analysis (LDA), or Partial Least Squares Discriminant Analysis (PLS-DA). On the other hand, class-modeling methods focus on the similarities among samples from the same class rather than the differences between classes. In class modeling, classes are modeled individually and independently, and a sample can be assigned to a given class, to more than one class, or to none of the classes. For a two-class problem (A and B), a sample could be assigned to A, to B, to A and B, or to neither A nor B. A typical example of a class-modeling technique is SIMCA (Soft Independent Modeling of Class Analogies). In beer science, classification methods are extensively applied to discriminate between geographical origins, to assess brand authenticity or beer style, to assess raw materials, and also to discriminate between fresh and aged beer [9,41,66]. The choice of a particular technique will depend on the nature of the problem at hand; that is, whether it is discriminant or class modeling (i.e., PLS-DA vs. SIMCA), whether it is linear or non-linear (i.e., LDA vs. ANN), or depending on the number and degree of correlation of the measured variables (i.e., LDA vs. PLS-DA).

Classification techniques can be applied in the beer production process in order to verify the authenticity of the product based on the raw material or the production process. Silva et al. [45] built PLS-DA models to distinguish lager beers based on the type of raw materials employed in the brewing process. The authors used NMR spectroscopy combined with chemometrics to discriminate lager beer samples according to their style and the raw material information provided on the label. The authors concluded that the approach adopted could be very useful when applied to a suitable set of samples. The models obtained had a prediction power higher than 90%, considering the raw material employed in the brewing processes. Vivian et al. [12] used PLS-DA to characterize markers of key production stages of the brewing process of a Brazilian craft brewery using electrospray ionization (ESI) high-resolution mass spectrometry (HRMS). The authors concluded that their approach allows a quick assessment of the process status before it is finished without the subjectiveness of sensorial analysis, thus preventing higher production costs, ensuring quality, and helping with the control of desirable features, such as flavor, foam stability, and drinkability. Gianetti et al. [60] evaluated the flavor profile from craft beers (unfiltered and unpasteurized) and industrial beers. The aim was to characterize craft beers to differentiate them from industrial mass-produced beers. PLS-DA was used to classify beers according to their different production methods. The results showed a good classification, both in calibration (96.2%) and cross-validation (94.2%), enabling a good separation between beer categories with high prediction accuracy (96.23%).

Several studies involving pattern recognition methods to monitor the aging process have been carried out. Linear techniques such as PCA and LDA were applied by Ghasemi-Varnamkhasti et al. [50] to characterize the change of aroma of alcoholic and non-alcoholic beers from the same brand during the aging process by using metal oxide semiconductor-based electronic nose. The results did not reveal clear discrimination among alcoholic aged beers, showing more stability of such types of beer compared with non-alcoholic aged beers. Rodrigues et al. [46] applied PCA and PLS-DA to NMR spectra to monitor the chemical changes occurring in a lager beer exposed to forced aging. Inspection of PLS-DA loadings and peak integration enabled the changing compounds to be identified and revealed the importance of well-known aging markers, as well as other relevant compounds.

Understanding and controlling the aging process of a beer remains a hard task for brewers. In a study by Ghasemi-Varnamkhasti et al. [47], the potential of NIR spectroscopy for the qualitative analysis of different types of beer during the aging process was measured. PCA, KNN, LDA, Stepwise LDA, Genetic Algorithms (GA) and Gram–Schmidt supervised orthogonalization (SELECT) were employed to characterize the aging phases as well as beer types. The results demonstrated that the computational tools were capable of discriminating and classifying the aged beers, showing high classification accuracies for all aging treatments.

Classification techniques have also been applied for authentication purposes. Tan et al. [68] discriminated Chinese lager beers produced by different manufacturers, with good accuracy. They used a data fusion approach by combining fluorescence, UV, and visible spectroscopies. LDA and PCA-LDA (LDA applied on the scores of a previous PCA on the data) were applied, showing a much better classification accuracy (79–87%) when compared with the classification models on the individual instrumental techniques (42–70%). Gordon et al. [64] used Mid-infrared (MIR) spectroscopy coupled with attenuated total reflectance (ATR) to classify different beer types (ale vs. lager and commercial vs. craft beer). PLS-DA was used to analyze and to discriminate the beer samples based on their infrared spectra. Correct classification rates of 100% were achieved in all cases, showing the capability of MIR spectroscopy combined with PLS-DA to classify beer samples according to style and production. Furthermore, dissolved gases in the beer products were shown not to interfere as overlapping artefacts in the analysis. The benefits of using MIR-ATR for rapid and detailed analysis coupled with multivariate analysis can be considered a valuable tool for researchers and brewers interested in quality control, traceability, and food adulteration. 

### 3.3. Multivariate Calibration

Multivariate calibration aims to create a mathematical model to predict properties of interest from instrumental measurements. Modeling can help control and optimize process performances, which are very useful to the beer production process. Moreover, to better understand the aging process, multivariate calibration can link chemical and sensory data [41]. The most commonly used multivariate calibration techniques are multiple linear regression (MLR), principal component regression (PCR), and partial least squares (PLS) regression. These calibration models need a thorough validation before they can be used to reliably predict system or product properties [9]. 

An example of the use of multivariate calibration was shown in Sturm et al. [34]. In this study, the authors investigated the dynamics in the drying behavior and quality development of hops using visual and environmental sensors combined with chemometrics. To better understand the dynamics of the drying process, a full array of visual sensors was integrated into a pilot scale drying system to investigate the color changes, Hop Storage Index, α acids and β acids, product and air temperature, and air humidity, throughout the drying process with different bulk weights and drying temperatures. PLSR was applied in combination with spectroscopy and hyperspectral imagining. The results showed that, besides bulk weight and temperature, harvesting conditions and specific air mass flow have a significant influence on both drying time and color changes of hops during drying in identical conditions.

Gagula et al. [67] created mathematical models using two partial least squares regression methods: polynomial regression (PLSR-PR) and response surface method (PLSR-RSM) to describe changes in beer properties during storage based on three measured properties: color, bitterness, and haze values. The samples used were lager beers packed in glass bottles and polyethylene terephthalate (PET) bottles and samples of malt beer in glass bottles. The authors concluded that PLSR-RSM models were more accurate when describing property changes for the lager and malt beer in glass bottles, while PLSR-PR was better for the lager beer in PET bottles. By comparing the samples, the models showed that beer packaging in PET bottles showed larger changes than lager beer in a glass bottle during the storage period. In contrast, both lager beer and malt beer showed great changes in different periods of storage.

Partial Least Squares (PLS) regression was used by Krebs et al. [58] to create a model to predict the palate fulness intensity in beers. In this article, the authors reported a chemometric analysis of 41 lager beers based on the evaluation of the analytical data of beer composition, palate fullness, and mouthfeel. Ethanol, original gravity, dynamic viscosity, nitrogen and β-glucan were analyzed. The macromolecular profile of the samples was analyzed and a sensory characterization was performed by certified panelists. The authors concluded that palate fullness and mouthfeel are key factors that determine the quality of lager beer and consumers’ acceptance, and that the prediction model can be used for a targeted design of palate fullness by weighting the influence factors. Calibration models can also be used to predict the organoleptic quality of the beer during the aging process. Hemp et al. [56] used an optical oxygen sensor to assess the level of residual oxygen in the headspace of bottled beers by monitoring the product over time before and after pasteurization. A sensory panel was also used to determine the effect of the residual oxygen on the sensory quality of the product. PLS-R was used to process the sensory data obtained by the 26 panelists. The results showed that the higher the oxygen level prior to pasteurization, the more negative the attributes associated with the sensory quality of the beer, especially those related to beer staling.

Another example of multivariate calibration was used to create a method for the retrospective determination of temperature based on the determination of carbonyl compounds determined by GC-MS. Čejka et al. [71] used three approaches: regression graph, multiple linear regression (MLR), and neural networks, to calculate the storage temperature of samples. In this study, 11 samples from the eight major Czech breweries were stored for 6 months at 0, 8, 20, and 30 °C. The MLR calculation used only 2-furfural as representative indicator of aging. The exponential dependency of 2-furfuralwith storage was converted to a linear dependency using a logarithmic transformation and a regression equation was created, with months and ln *_c_*(furfural) as the input variables and storage temperature as the output variable. The uncertainty of the final predictions was 5 °C.

## 4. Conclusions

Despite being extensively studied, beer quality is still a challenge for the beer industry. Maintaining the quality and stability of the beer during its shelf life requires monitoring the entire beer production process from the field to the final consumer. Chemometric techniques can help us to better understand both the beer production and the aging processes, by extracting useful information from large and complex brewing datasets.

Further studies involving a comprehensive view of beer stability, combining analytical techniques and chemometric solutions, could help us to better understand the process and make corrections in real time, avoiding losses, saving time, and guaranteeing a higher-quality product.

In conclusion, unsupervised and supervised chemometric methods are powerful tools that have proved to be quite useful for monitoring the beer production and aging processes. Unsupervised exploratory methods allowed us to obtain the first insights about the data, and supervised approaches (discriminant analysis and multivariate calibration) revealed the underlying correlations between sensory and chemical changes in beer during the process and shelf life of the product.

## Figures and Tables

**Table 1 foods-11-02037-t001:** Malt type, color, and general characteristics [10,18].

Malt Types	Color SRM ^1^	Color Description	Organoleptic Characteristics
**Base Malts**			
Pilsner	1.2–2	Very Pale	Little green, with the smell and taste of fresh wort.
Pale	1.6–2.8	Light colored	Deeper malt aroma than Pilsner.
Pale Ale	2.7–3.8	Darker than standards pale malts	Not excessively pronounced malt aroma, with notes of biscuit or toast.
Vienna Malt	2.5–4.0	Imparts a rich orange color to beer	Slightly toasty and nutty
Melanoidin Malt	17–25		Sweet honey-like flavor.
Munich	3–20	Covers a broad range of colors	Malty profile.
**Specialty Malts**			
**Caramel Malts**			
Special Glassy (Carapils)	1–12		Add body and impart sweetness to beer
Caramel/Crystal	10–200	Can imply significant color differences depending on the method of manufacture.	Can imply significant aroma differences depending on the method of manufacture.
**Roasted Malts**			
Biscuit	20–30		Bread crust, nutty, and toasted aromas. Dry finish.
Amber	20–36		Nutty, biscuit, toffee taste.
Brown	40–150	Darker than Amber.	Nutty, biscuit, toffee taste.
Chocolate	350–500	Dark color.	Treacle and chocolate aromas. Present dray and ashy aspects.
Black	435–550		Bitter, dry, and burnt aromas.
Roasted	300–650		Smoky, coffee, chocolate, and roast aromas.

^1^—The Standard Reference Method, abbreviated SRM, is the color system used by brewers to specify finished beer and malt color.

**Table 2 foods-11-02037-t002:** Scientific literature (2012–2021) correlated to the chemometric techniques applied to beer quality and stability.

Aim of the Study	Year	Analytical Techniques	Chemometric Techniques	Reference
Proposal of a methodology fast non-destructive metabolomic characterization of beer exploring the compositional profile of the product.	2021	NMR spectroscopy	PCA MCR (Multivariate Curve resolution)	[57]
Evaluation of the factors that influence the perception of the intensity of palate fullness and selected descriptors of mouthfeel in fresh lager beer.	2021	Physical chemical parameters Macromolecular characterization Sensory panel	HCA PLS	[58]
Understand the changes during the drying process to optimize the process, improving the process performance and the quality of the product.	2020	Hyperspectral imaging	PLSR	[34]
Metabolomic profiling of beers to discriminate craft and industrial products.	2020	NMR spectroscopy	PCA PLS-DA	[59]
Build and test a model capable of estimating the quality of beer.	2019	Sensory panel	The model was created using Curve Fitting Toolbox in Matlab	[40]
Differentiate Brazilian lager beers by styles employing NMR spectroscopy combined with chemometric approach.	2019	H NMR	PCA PLSDA SIMCA	[45]
Characterize the craft beers to differentiate them from the other competing and lower-quality products.	2019	GC-MS	PLS-DA LDA	[60]
Multivariate analysis as a tool to discriminate and characterize differences in barrel diverse-aged beers using volatile fingerprinting.	2019	GC-MS	PCA	[61]
Understand if there would be metabolite differences among six commercial barley sources and if this difference is reflected in the chemistry and in the sensory attributes of beer.	2018	UHPLC-MS HILIC-MS GC-MS ICP-MS Sensory analysis	PCA	[23]
Compounds behavior in natural and forced aging—recommendations as to how prediction by forced aging should be used.	2018	GC-O GC-MS Sensory panel	PCA	[62]
Beer volatile terpenic compounds.	2018	HSPME-MS GC × GC-TOF-MS	HCA	[63]
Traceability, quality control, and food adulteration.	2018	Mir spectroscopy coupled with attenuated total reflectance (ATR)	PCA PLS-DA	[64]
Method optimization for volatile aroma profiling of beer.	2017	GC × GC-TOF-MS	PCA HCA	[65]
Characterization of brewing process—“Processomics”.	2016	Electro spray ionization-Mass Spectrometry (ESI-MS)	PLS-DA	[12]
Differentiation between beers according to their price market.	2016	Paper spray mass spectrometry (PS-MS)	PLS-DA	[66]
Create mathematical models that can be used during the measurement of beer shelf life.	2016	Physical chemical parameters Haze	PLSR-PR PLSR-RSM	[67]
Developing accelerated model to evaluate brewing techniques that affect flavor stability using metabolomics on non-volatile compounds in beer.	2016	UPLC-MS	PCA	[7]
Study volatile profiles and characterize odor-active compounds of brewing barley in order to determine the variability of the aroma composition among different brewing barley cultivars.	2015	GC-MS	PCA Hierarchical Clustering	[52]
Propose a methodology for determining the start of the period of time in which beer fresh features start to change.	2015	GC-MS	PCA	[53]
Using data fusion to establish a model to classify Chinese lager beer according to the manufacturer.	2015	Fluorescence/UV/Visible spectroscopies	PCA LDA	[68]
Monitoring the aging process in alcoholic and non-alcoholic beers.	2014	NIR	PCA KNN LDA StepLDA GA SELECT	[47]
Investigate the volatile metabolomic profile of raw materials used in beer.	2014	HS-SPME GC-qMS	PCA SLDA	[69]
Determine the effectiveness of incorporating an oxygen sensor into lager beer bottles and predicting the sensory quality of the beer with respect to the oxidation and staling.	2013	Optic oxygen sensors Sensory panel	PLSR	[56]
Clarify the aroma compounds affecting the various hop aroma characteristics, using beer prepared with different hop varieties.	2013	GC × GC-TOF-MS	PCA	[70]
Development of a method for retrospective determination of temperature conditions to which beer had been exposed	2013	GC-MS Sensory panel	MLR ANN	[71]

## Data Availability

No new data were created or analyzed in this study. Data sharing is not applicable to this article.

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
