# Peer review of "An Overview of the Application of Multivariate Analysis to the Evaluation of Beer Sensory Quality and Shelf-Life Stability"

_foods, 2022, doi:10.3390/foods11142037_

Round 1

Reviewer 1 Report

Comments have been provided in the manuscript for the authors consideration. The table should be revised.

Reviewer 2 Report

Paper reads well with extensive review on materials but very brief section on chemometrics.

Table 3 is hard to read, please add a separate line for each study.

The authors mentioned machine learning techniques such as ANN but it is not on the Table summarised?

The authors provided an extensive review on the techniques that has been used. However, no extensive comparisoms was done. For example, PLSDA can be used to classify beers, but so does other classification clustering technique. Can the authors attempt to explain this?

Perhaps the authors should attempt to subhead their sections in terms of the nature of techniques that is used. For example, classification/clustering, regressions, etc.

Reviewer 3 Report

The topic of the manuscript is very interesting but there are a lot of inacurracies in manuscript that decrease its quality. 

First, lautering is carried out in mash tun only in craft breweries (ln 77).

Ln 78 is correct only if spent grains are sparged with water.

Ln 83 not concentration of sugars, but wort concentration

Ln 85-86 wort has to be not only cooled but also aerated

Ln 92-101 It will be good to mantioned that some beers are filtered

ln 127 to grown -> to grow

ln 256-262 you have replaced the fermentation temperatures of ale and lager strains

Round 2

Reviewer 2 Report

Comments were addressed accordingly

Author Response

We are very grateful for the time spent by the reviewer and the useful comments and suggestions.

Reviewer 3 Report

The authors have taken into account all the reviewers' notes so the quality of the paper has improved considerably.

Author Response

We are very grateful for the time spent by the reviewer and the useful comments and suggestions